# Liquid Biopsy, ctDNA Diagnosis through NGS

**DOI:** 10.3390/life11090890

**Published:** 2021-08-28

**Authors:** Chen Lin, Xuzhu Liu, Bingyi Zheng, Rongqin Ke, Chi-Meng Tzeng

**Affiliations:** 1School of Medicine, Huaqiao University, Quanzhou 362021, China; chen.lin@hqu.edu.cn (C.L.); liuxuzhu@stu.hqu.edu.cn (X.L.); 2Translational Medicine Research Center, School of Pharmaceutical Sciences, Xiamen University, Xiamen 361102, China; leozheng2021@126.com; 3Xiamen Key Laboratory of Cancer Cell Theranostics and Clinical Translation, Xiamen 361102, China

**Keywords:** liquid biopsy, ctDNA, next-generation sequencing, mutation, biomarkers

## Abstract

Liquid biopsy with circulating tumor DNA (ctDNA) profiling by next-generation sequencing holds great promise to revolutionize clinical oncology. It relies on the basis that ctDNA represents the real-time status of the tumor genome which contains information of genetic alterations. Compared to tissue biopsy, liquid biopsy possesses great advantages such as a less demanding procedure, minimal invasion, ease of frequent sampling, and less sampling bias. Next-generation sequencing (NGS) methods have come to a point that both the cost and performance are suitable for clinical diagnosis. Thus, profiling ctDNA by NGS technologies is becoming more and more popular since it can be applied in the whole process of cancer diagnosis and management. Further developments of liquid biopsy ctDNA testing will be beneficial for cancer patients, paving the way for precision medicine. In conclusion, profiling ctDNA with NGS for cancer diagnosis is both biologically sound and technically convenient.

## 1. Introduction

It is now well known that a tumor can shed cancer cells, extracellular vesicles, proteins and nucleic acids into the peripheral blood [1]. These tumor-derived substances possess great potential as therapeutic targets or diagnostics biomarkers (Figure 1). The idea of liquid biopsy is to sample the biomarkers found in the non-solid biological tissue for analysis of disease status. Different biological substances such as circulating tumor cells (CTCs), cell-free nucleic acids, exosomes, or proteins can be sampled and analyzed in various body fluids such as blood or urine [2,3]. In particular, the blood is the most widely used sampling source of liquid biopsy due to the rich information it carries and because it is easy to access [4]. Among these blood-borne biomarkers, circulating tumor DNA (ctDNA) has gained a tremendous amount of interest in recent years. In this review, we discuss the biology, clinical utilities, and applications of ctDNA as biomarkers, emphasizing the technologies for their detection.

## 2. Biology of ctDNA

ctDNA is shed into the peripheral blood by tumor cells in cancer patients. However, the biological mechanisms of how ctDNA enters and presents in the blood circulation have not yet been fully elucidated [5]. Generally, it is widely accepted that apoptosis and necrosis of tumor cells are the main sources of ctDNA. Studies have revealed that ctDNAs are generally more fragmented than nonmutant cell-free DNA (cfDNA), with a maximum enrichment of fragments between 90 and 150 bp [6] while the half-life of ctDNA ranged from only 15 minutes to a few hours [7]. Although it is challenging to preserve and use ctDNA as a biomarker because of the short half-life time window, it may reflect a snap-shot of the tumor genome status, providing a way to monitor the disease. However, ctDNA represents only a small part of cfDNA in the circulation. Several biological processes are known to release cfDNA, including phagocytosis, active secretion, neutrophil release, excision repair, and pyroptosis [8]. Thus, discriminating ctDNA from the normal cfDNA is vital to precisely reflect the disease status. Since ctDNA is derived from malignant tumor cells, they carry the tumor-specific abnormality that is not present in the normal cfDNA. The tumor-specific abnormalities that ctDNA possesses are inherited from its tumor origins, such as single-nucleotide mutations, epigenetics changes, and copy number variations [9]. These molecular features can then be used as specific biomarkers for cancer monitoring and management references.

Apart from tumor-specific molecular characteristics, studies have also found that the levels of ctDNA in advanced stage cancers such as breast, colorectal, pancreatic, and gastroesophageal cancers are higher than in the early stage of diseases [10]. However, the amount of ctDNA is not simply associated with the tumor size, but varies in different types of cancer [11,12]. Furthermore, ctDNA can both derive from primary tumor and metastases, thus is associated with the pathological status of the disease [13]. Because of the different origins of ctDNA, liquid biopsy together with ctDNA analysis may be a better representation of tumor genomic diversity compared to biopsy from a single solid tumor sample [14]. In other words, ctDNA reflects a better overview of the most up-to-date status of the tumor genome, making it superior to tissue biopsy analysis for cancer patient diagnosis and management. Circulating cell-free RNA (ccfRNA), mainly including mRNA and miRNA, is another type of cell-free nucleic acid that can be found in the blood stream, and also possesses potential to become a biomarker for cancer [15]. Although ccfRNA shares many similar characteristics to ctDNA, there are more challenges when using them as cancer biomarkers [16]. Thus, more studies and technical advances are still needed before ccfRNA can become as popular as ctDNA in cancer diagnosis. Therefore, we focus on the discussion of ctDNA in this review.

## 3. Next Generation Sequencing Technologies for ctDNA Detection

Because the ctDNA represents only a small part of the cfDNA, it is important to utilize the tumor-specific alterations to distinguish themselves from their wild-type counterparts. Currently, most methods using ctDNA as cancer biomarkers focus on mutation de tection [17]. Mutations in ctDNA can be detected either by PCR or NGS [18,19]. However, PCR approaches only detect known mutations in certain genes. Patients without those mutations will be left out, limiting their applications as a generic diagnosis technique for ctDNA analysis. NGS approaches, on the other hand, cover a broader range of mutations by examining the whole sequences of genes of interest. In this review, we focus on discussing the NGS approaches for ctDNA profiling.

NGS for ctDNA analysis can be targeted or untargeted as well [20]. Targeted approaches for ctDNA profiling usually sequence dozens of genes to hundreds of genes or even the whole exome. Because of relatively low throughput, high sensitivity can then be achieved by deep sequencing specific regions of interest that cover clinically relevant mutations. To enrich targets, either multiplex PCR or hybridization capture strategies are ap plied to ensure maximal amplification of target gene fragments. Due to better specificity and sensitivity, targeted sequencing is more suitable for clinical diagnosis. In the case of untargeted approaches, no enrichment step is performed, thus sequencing the whole genome. Although whole-genome sequencing scarifies the sequencing depth, it can be used to discover new genomic aberrations related to patient prognosis or treatment strategies, making it suitable for basic biomedical research.

For the sequencing step, the cost, accuracy, and speed are all key factors that need to be taken into consideration for ctDNA profiling. It is particularly true for diagnosis purposes. Under this circumstance, there are currently several platforms more suitable for ctDNA sequencing, including Illumina’s sequencing-by-synthesis (SBS), Thermo Fisher Scientific’s Ion Torrent, and the nanopore sequencing from Oxford Nanopore Technologies [21,22]. Illumina’s sequencing platform is dominating due to its high throughput and high accuracy. The Ion Torrent semiconductor sequencer is lower in throughput but faster in the run speed, making it advantageous when speed is required. During the past decade, nanopore sequencing has been improving in its accuracy but is still lower than the other mainstream sequencers. However, it is advantageous in the convenient library preparation and sequencing procedure. Thus, in the case when the requirement of accuracy is not high, such as profiling copy number variations, nanopore sequencing can be used for ctDNA analysis [23]. After generation of raw sequencing data, applying the right bioinformatic algorithms is also crucial for mutation detection. A review that summarized bioinformatics tools for ctDNA analysis can be found here [24]. There are also databases that can be used to help with the interpretation of the liquid biopsy data. liqDB is a database for liquid biopsy small RNA sequencing profiles that provide users with meaningful information to guide their small RNA liquid biopsy research and to overcome technical and conceptual problems [25]. ctcRbase is a database that can be used for query and browse gene expression data of circulating tumor cells/microemboli [26].

The key for precisely identifying ctDNA from a large amount of background cfDNA is to correctly detect the alterations of ctDNA compared to normal cfDNA and single nucleotide mutation is the most commonly detected alteration [27]. Technically, in the context of ctDNA sequencing, the sensitivity of detecting the so-called mutant allele frequency (MAF) or variant allele frequency (VAF) is often used to evaluate the performance of an NGS ctDNA profiling assay [13,28]. MAF or VAF describes the proportion of DNA molecules containing a mutation over the total number of molecules containing the same allele. Thus, they reflect the amount of ctDNAs when referring to the cfDNAs that contain tumor-specific mutant alleles. Therefore, the lower the MAF can be detected, the more sensitive an NGS assay is for ctDNA analysis.

## 4. Strategies for True Mutation Detection Using NGS

Because ctDNA is mainly distinguished from normal cfDNA by tumor-specific mutations, it is thus critical to identify a true mutation for sensitive ctDNA detection. Plus, the low levels of ctDNA presented in the bloodstream also warrant highly sensitive and specific methods that can detect and quantify mutant alleles in cfDNA. This can be achieved both by experimental designs and bioinformatics algorithms. Some of these strategies are presented below to elucidate the general principles.

Tagged-amplicon deep sequencing (TAm-Seq) introduces a two-step amplification strategy to first amplify regions of interest by multiplex PCR using specifically designed primers at low concentrations to avoid inter-primer interactions, followed by using a microfluidic system in a second amplification step to amplify individual target regions using the amplicons from the first step [29]. Sequencing adaptors were attached to the amplification products and unique molecular identifiers were used to tag different samples by another PCR step. Tam-seq was able to identify mutations of MAF as low as 2% with sensitivity and specificity over 97% [29]. The enhanced TAm-Seq (eTAm-Seq) can detect MAF as low as 0.25% with a sensitivity of 94%. It also utilizes revised bioinformatics analysis to identify single-nucleotide variants (SNVs), short insertions/deletions (indels) and copy number variants (CNVs) [30].

Unique molecular identifiers (UMIs) are often used to barcode individual templates so that the amplicons from different templates can be identified [31]. UMIs are normally short DNA fragments that are constituted by random nucleotide bases. The idea is that for a DNA sequence of N bases, there will be 4N different types of barcodes that can be used to label different templates before amplification. If a mutation does not appear in most of the same UMI-connected sequences, it is likely to be an error introduced by amplification or the sequencing reaction. By using this strategy, the Safe-SeqS reduces the sequencing errors by at least 70-fold and has a sensitivity as high as ~98% for detecting tumor-specific mutations [32].

A combination of optimized library preparation and bioinformatics algorithm is often used to achieve better performance of NGS assays for low-abundance sequence aberration in ctDNA analysis. Target enrichment of DNA fragments containing recurrent mutations combined with deep sequencing can also improve the sensitivity of mutation detection. CAncer Personalized Profiling by deep Sequencing (CAPP-Seq) is such an NGS assay that combines a target-selection library preparation approach with specialized bioinformatics workflow [33]. It generates a library of selectors that can specifically bind to recurrently mutated genomic regions for target enrichment. Combined with dedicated bioinformatics analysis, CAPP-Seq can detect MAF ~ 0.02% with a sensitivity of nearly 100% among stage II-IV NSCLC patients [33]. Targeted error correction sequencing (TEC-Seq) is another example of experimental and bioinformatic optimizations to improve assay performance, including optimized target capture and library preparation, maximizing the representation of unique cfDNA both using different mapping positions and molecular barcodes, redundant sequencing, filtering of mapping and sequencing artefacts, and identifying and removing normal cell proliferation alterations [34]. Target capturing a panel of genes and deep sequencing (~30,000×) of the captured DNA fragments is performed to ensure high sensitivity and specificity. It was able to detect somatic mutations in the plasma of 71%, 59%, 59% and 68%, respectively in colorectal, breast, lung, or ovarian cancer patients with stage I or II diseases, demonstrating very promising results for broad cancer diagnosis at early stages.

## 5. Detection of ctDNA Mutations for Early Cancer Diagnosis

Although early diagnosis of cancer is the key to reduce morbidity and mortality of cancer, it is still challenging due to technical limitations, especially for the broad diagnosis of different types of cancer. The postulation of using ctDNA for early cancer diagnosis is that tumors will release fragmented nucleic acids into the circulating bloodstream before they can be visualized by other diagnosis approaches or cause perceptible symptoms. Therefore, early diagnosis of cancer by analyzing ctDNA alterations requires both sensitivity and specificity. If the method can also determine the origin of the ctDNA, it is likely to be an ideal method. However, this demands that the ctDNA detection technique identifies tumor-specific mutations from background cfDNA and their normal alteration in a minute amount of DNA fragments.

Many cancer patients are asymptomatic at the early stage of diseases. For example, pancreatic cancer is a very aggressive disease, but most patients are diagnosed at the late stage with advanced metastasis. Additionally, due to the lack of therapeutic options for advanced disease, pancreatic cancer patients have extremely poor 5-year survival. However, math ematical modeling of the mutation acquisition rate suggests that the development of pancre atic cancer is extremely slow in that an 11.7-year period from acquisition of the initiating mutation to full transformation in a pancreatic cell, and another 6.8 years are needed to develop the first metastatic subclone. Thus, early diagnosis is the key to improve the prog nosis of pancreatic cancer patients [35]. Mutation detection of ctDNA can potentially help improve patient outcomes. KRAS, CDKN2A, TP53, and SMAD4 are known to be commonly altered in pancreatic cancer [36] while KRAS mutation has been proposed as a biomarker in cfDNA for the detection of PDAC [35]. However, in early-stage malignant dis ease and in some metastatic cancers, ctDNA may be extremely rare in total cfDNA (0.01% or less) [33,37]. Thus, there are still technical challenges to develop ctDNA testing assays for early diagnosis of pancreatic cancer.

CancerSEEK is a blood test that aims for early diagnosis of different types of cancer [38,39]. It is also based on ctDNA mutation detection using NGS but accompanied with protein biomarker detection to assist the diagnosis when ctDNA is low or absent. Studies have found that a major fraction of early-stage tumors do not release detectable amounts of ctDNA, even when very sensitive methods are applied for the detection [40,41], indicating technology combination may be a good option to avoid false negatives.

Liquid biopsy for ctDNA testing seems to give promising biomarkers for asymptomatic early cancer diagnosis. However, theoretical calculations based on reported tumor measures show that when the fraction of tumor DNA falls below 0.01% of the total cfDNA, drawing 10 mL of blood (normally contains 4 mL of plasma) for ctDNA testing will likely to contain less than one cancer genome [42]. In such a low level of ctDNA, early diagnosis of cancer is very unlikely to be achieved. Therefore, improvements in technologies are still needed before sensitive assays for early cancer diagnosis can be generally applied in clinical settings. A recent study showed that detection of ctDNA via deep methylation sequencing aided by machine learning identified nearly twice as many patients with can cer as those identified by ultradeep mutation sequencing analysis [43]. It also enabled the detection of tumor-derived signals at dilution factors as low as 1 in 10,000, showing its great potential for being used as an ultrasensitive method for early cancer diagnosis [43]. Furthermore, other studies demonstrated that the tissue-of-origin of ctDNA can be tracked by profiling the methylation patterns, demonstrating its potential for early diagnosis of asymptomatic cancer patients [44,45]. However, a study also showed that simply combining ctDNA methylation and mutation profiling did not improve the diagnosis performance [46], indicating further studies are still needed to implement the diagnosis values and strategies of combined mutation and methylation profiling for tumor early diagnosis.

## 6. Detection of Therapeutically Targetable Mutations in ctDNA

Genetic mutations found in the genomic DNA that can be treated with available anticancer therapeutics or the target of developing new drugs are called targetable mutations or actionable mutations. Targetable mutations are clinically relevant because it is the ultimate goal to guide treatments for the detection of mutations in ctDNA (Table 1). Many of these targetable mutations, such as mutations in KRAS, EGFR, BRAF, and PIK3CA, are often detected by PCR due to its low cost, fast speed, and ease of operation [47]. Like the other PCR tests, only detecting known mutations and a relatively small number of targets limit their applications to certain patients and diseases. Compared to PCR-based approaches, NGS can detect more genes simultaneously and covers more different types of alterations in genes, not only point mutations, but also insertion or deletion, amplification and microsatellite instability [48], as well as gene fusions and translocations that may be targetable in different types of cancer [49,50,51,52]. Now, the FDA has already approved two NGS- based liquid biopsy ctDNA testing assays, the Guardant360 CDx assay and the FoundationOne CDx assay which can sequence 55 genes and 311 genes respectively [53]. Both of these two assays rely on hybridization to capture cfDNA fragments for genes of interest from the plasma for the following sequencing analysis. There are also many other companies developing NGS ctDNA testing assays. Together with increased depth of coverage as well as decreasing cost, NGS may be the dominant technology for the future liquid biopsy ctDNA testing assays [54,55].

Tumor-specific mutations in the coding region of genes can produce mutant proteins which are not present in normal cells. These mutant proteins, called neoantigens, can activate the immune system to attack the cancer cells [75]. Neoantigens can either be commonly shared within patients or unique to specific patients. The former can be used to produce broad-spectrum cancer vaccines to treat patients with the same type of gene mutations [62], and the latter will only be able to help produce personalized vaccines for individuals carrying that type of mutation [76]. Many studies and clinical trials are trying to use shared mutations as candidates, aiming for developing broad-spectrum neoantigen therapeutic strategies. Thus, ctDNA liquid biopsy could serve as companion diagnostics for this kind of treatment by analyzing the neoantigen-related genetic alterations. Although the evolutionary dynamics of tumor-associated neoantigens can be used to evaluate drug sensitivity and resistance to the immune checkpoint blockade (ICB), it is difficult to track neoantigens by ctDNA sequencing using commercial gene panels with predefined gene sets due to the highly heterogeneous mutations among patients. Jia, et al., therefore developed individually customized panels (ICP) targeting predicted neoantigens for personalized ctDNA sequencing to track neoantigen evolution [77]. They demonstrated that their approach was able to identify patients likely to have favorable outcomes following ICB treatment, as well as to identify responsiveness and to indicate the survival benefit of ICB, demonstrating ICP-based ctDNA sequencing provides superior coverage to longitudinally track predicted dominant neoantigens. This kind of approach may hold great potential to identify targetable mutations more effectively in ctDNA liquid biopsy. Although most of the ctDNA NGS assays focus on the detection of known mutations, it is also possible to identify de novo mutations. To validate if such de novo mutations are true mutations, references such as paired normal samples can be used to exclude normal variants [78]. The functions of such de novo mutations can be predicted by methods such as in silico prediction [79], thus indicating if they are actionable or not. Because the mutation characteristics in the tumor and ctDNA are not static [80], continuous monitoring is important to reveal the status of disease.

## 7. Detection of ctDNA Mutations for Cancer Monitoring and Assessment of Minimal Residual Disease

To implement precision methods for cancer therapeutic strategies, there is an increasing interest in disease prognosis using genomic alterations [81]. ctDNA from liquid biopsy has great potential as an ideal biomarker for these kinds of applications due to its advantages both biologically and technically. Applications for disease monitoring have already been used in different types of cancers. ctDNA profiling can be used for treatment assessment, for example, in hepatocellular carcinoma (HCC) patients treated with lenvatinib (LEN) [82]. By monitoring the level changes of variant allele frequencies (VAFs) in ctDNA, the study showed that somatic alterations could be detected in the majority of advanced HCC patients and that the reduction of VAFs is associated with longer progression-free survival [82]. Compared to conventional protein biomarker serum α-fetoprotein level, the specificity and sensitivity of the reduction of mean VAF for predicting partial response were higher (0.67 and 1.0 v.s. 0.10, and 0.93) [82]. Lung and bladder cancer patients treated with durvalumab that showed a reduction in VAF at 6 weeks had a greater reduction in tumor volume, with longer progression-free and overall survival [83].

The efficacy of neoadjuvant chemotherapy (NAC) can be evaluated by ctDNA profiling. Preliminary results of a study showed that combined profiling of ctDNA by NGS with functional tumor volume (FTV) by magnetic resonance imaging (MRI) improves the prediction of metastatic recurrence and death of patients treated with NAC [84]. They found that the levels of ctDNA (mean tumor molecules/mL plasma) were correlated with FTV at all time points, and patients with ctDNA-positivity after NAC showed an increased risk of metastatic recurrence and death. Another study showed that breast cancer patients without ctDNA mutation-positive blood after NAC showed a superior primary tumor decrease and lymph involvement [85]. These studies show that monitoring of ctDNA is a potentially effective way for the evaluation of NAC efficacy and patient recurrence.

Tumor evolution can also be accessed by NGS ctDNA profiling. A study in non-small cell lung cancer (NSCLC) patients demonstrated that more mutations were detected in targetable genes by ctDNA than tissue biopsy samples [86]. Another study exploiting dynamic ctDNA analysis to monitor genetic evolutions during ensartinib treatment in NSCLC showed that TP53 mutations promoted tumor evolution and accelerated occurrence of resistance, thus indicating TP53 mutations at baseline were independently correlated with worse clinical outcomes [87]. Deep targeted and whole-exome sequencing of ctDNA in metastatic castration-resistant prostate cancer patients for comparison of base-line and posttreatment showed that the dominant androgen receptor (AR) genotype continues to evolve during sequential lines of AR inhibition and drives acquired resistance [88].

Minimal residual disease or measurable residual disease (MRD) is a term used for describing the presence of a small number of tumor cells in the body after treatment. MRD is responsible for the relapse of cancer [89,90]. Thus, better assessment of MRD is key for cancer management [91]. The number of cancer cells that remain in a patient’s body after effective treatment is usually so small that they are undetectable by conventional methods. Although MRD does not cause symptoms for a while, it may later lead to a recurrence of cancer. NGS-based ctDNA detection method is superior to traditional clinical or imaging methods, which can be used to evaluate the level of MRD in patients and has high sensitivity and specificity in predicting the risk of disease recurrence [92]. This will enable doctors to dynamically monitor and confirm remission of the disease, detect early signs of recurrence, and start treatment as early as possible to effectively manage cancer.

## 8. Conclusions

Because new generation sequencing is high-throughput, sensitive, specific and relatively cost-efficient nowadays, mutation profiling of ctDNA by NGS technologies is now showing its advantages over other techniques. It has already been demonstrated by many studies and commercial services that, NGS-based ctDNA profiling can potentially help to detect cancer early, effectively identify actionable mutations, as well as to help with the prognosis of cancer patient outcomes. Thus, NGS ctDNA profiling is already transforming molecular oncology to precision medicine. With technical advancements alongside price reduction, it can be foreseen that the NGS-based liquid biopsy for ctDNA profiling will accelerate its pace to be widely adopted in routine clinical diagnosis.

## Figures and Tables

**Figure 1 life-11-00890-f001:**
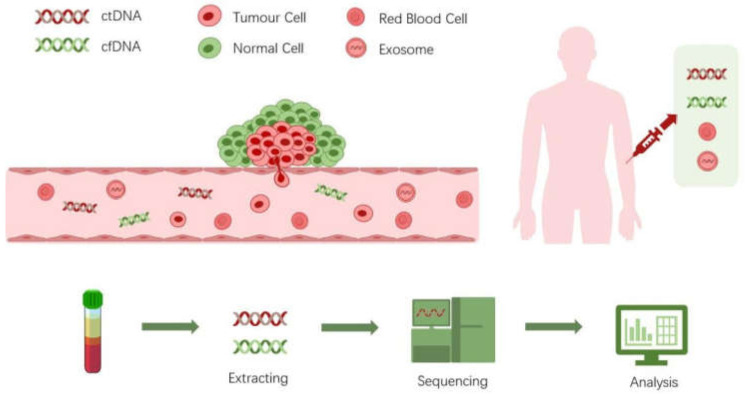
Deciphering of liquid biopsy.

**Table 1 life-11-00890-t001:** Targetable mutations in ctDNA.

Cancer	Gene	Mutation	Clinical Implication	Clinical Trials or Related Studies
Breast cancer	ESR1	D538G/Y537S/Y537C/Y538G	Biomarker of tumor evolution and prognostic impact	[56]
	PIK3CA	E545K/E542K/H1047R	Biomarker of tumor evolution and prognostic impact	[57]
Cervical cancer	ERBB2/HER2	S310F/V104M	Biomarker of tumor evolution and prognostic impact	[58]
	PIK3CA	R88Q	Biomarker of tumor evolution and prognostic impact	[59]
Colorectal cancer	KRAS	G12C/G12V/G12D/G12A/G13D/G12R	Biomarker of tumor evolution and prognostic impact	NCT04117087
	PIK3CA	E545K	Biomarker of tumor evolution and prognostic impact	[60]
Epithelial cancers	TP53	R175H/R249S/R273C/R273H	Potential targets for immunotherapy	[61]
Gliomas	IDH1	R132H	Biomarker of tumor evolution and prognostic impact	NCT 02454634
Kidney renal papillary cell carcinoma	MET	M1250T	Potential target for immunotherapy	[62]
Lung cancer	ERBB2	Internal tandem duplication	Augment CD4 + neoantigen-specific T cells	[63]
	KRAS	G12D	T cells target mutated antigen	[64]
	KRAS	G12V	Potential target for immunotherapy	[63,65]
Non-small cell lung cancer	CTNNB1	S37F/S45F/S45P/T41A	Biomarker of tumor evolution	NCT 03953235
	ERBB2/HER2	Y772_A775dup	Biomarker of tumor evolution and prognostic impact	NCT 03953235
	EGFR	T790M/C797S	Resistance of EGFR-TKIs treatment	[66]
	KRAS	G12C	Biomarker of tumor evolution	NCT 04685135
Lung adenocarcinomas	EGFR	L858R/E746_A750del	Prognostic impact	[67]
	EGFR	L858R	Potential target for immunotherapy	[65]
Lung Squamous cell carcinomas	EGFR	G719A	Assessment of neoantigen load and recurrence	[68]
	PIK3CA	E542K	Assessment of neoantigen load and recurrence	[68]
	TP53	V157F/G154V/R175G/P278A	Assessment of neoantigen load and recurrence	[68]
Melanoma	BRAF	V600E/V600K	Immunotherapy treatment monitoring	NCT 04364230
	NRAS	Q61K/Q61R/Q61L/Q61H	Early biomarker of tumor evolution	[69]
Ovarian cancer	TP53	R175H/C176S/R196P/R273H	Biomarker of tumor evolution and prognostic impact	[29]
Pancreatic cancer	KRAS	G12V	Biomarker of tumor evolution and prognostic impact	NCT 04146298
	KRAS	G12D	Biomarker of tumor evolution and Potential drug targets	NCT 03608631
	KRAS	G12R	Biomarker of tumor evolution and prognostic impact	[70]
Prostate cancer	AR	L720H/H874Y/T877A/T878A	Resistance of CYP17 inhibitor treatment	[71]
Pan-cancer	CTNNB1	S45F	Potential target for immunotherapy	[72]
Lymphoma	MYD88	L265P	Biomarker of tumor evolution and prognostic impact	[73]
	VDJ	rearrangement	Biomarker of tumor evolution and prognostic impact	[74]
Other solid tumor	BRAF	G466V	Biomarker of tumor evolution	NCT 03953235
	KRAS	G12C/G12D/G12V/G13D/Q61H/Q61K/Q61L/Q61R	Biomarker of tumor evolution	NCT 03953235
	TP53	K132E/K132N/R213L/R249M/S127Y	Biomarker of tumor evolution	NCT 03953235

## Data Availability

Data supporting the present findings are available from the corresponding authors upon reasonable request.

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
