# Peer review of "Liquid Biopsy, ctDNA Diagnosis through NGS"

_life, 2021, doi:10.3390/life11090890_

Round 1

Reviewer 1 Report

Authors presented liquid biopsy in terms of ctDNA by NGS, next-gen sequencing. They covered most of the topics related to enrichment methods, mutated genes, target variants, and mutant allele frequencies.

Here are some points that will be very helpful if authors add more paragraphs.

  1. there are no description on ctDNA methylation. Many researchers and companies seek to find ctDNA methylation patterns to pre-diagnose cancers. Sometimes, it could be use for finding tissue of origin if patients are not showing any symptoms.
  2. It would be helpful if authors can add how readers can acess bioinformatics software and how to use it.
  3. Many companies made a kit or methods by panel sequencing, targeting some cancer-related genes. In this case, one could find novel somatic mutations. Please describe more how we could deal with novel somatic mutation or rare somatic mutation in terms of diagnosis.

Author Response

Authors presented liquid biopsy in terms of ctDNA by NGS, next-gen sequencing. They covered most of the topics related to enrichment methods, mutated genes, target variants, and mutant allele frequencies.

Here are some points that will be very helpful if authors add more paragraphs.

  1. there are no description on ctDNA methylation. Many researchers and companies seek to find ctDNA methylation patterns to pre-diagnose cancers. Sometimes, it could be used for finding tissue of origin if patients are not showing any symptoms.

Reply: Indeed, methylation as an important epigenetic alteration also plays an important role in cancer. Most interestingly, it is shown that the methylation patterns can be used to track the origin of ctDNA. Therefore, we have added the descriptions on ctDNA methylation. On page 5, in the last part of the second paragraph counting from the bottom. we added “A recent study showed that detection of ctDNA via deep methylation… …strategies of combined mutation and methylation profiling for tumor early diagnosis”.

  1. It would be helpful if authors can add how readers can access bioinformatics software and how to use it.

Reply: Bioinformatic software is very important for interpreting the structure and function of the ctDNA, cfDNA, CTCs. We provided a review that give a better introduction on the bioinformatics tools for ctDNA analysis: Chen, S.; Liu, M.; Zhou, Y. Bioinformatics analysis for cell-free tumor DNA sequencing data. In Methods in Molecular Biology; 2018. Moreover, we also introduce two databases that can be used to assist data interpretation of liquid biopsy NGS. liqDB (https://bioinfo5.ugr.es/liqdb), a small-RNAseq knowledge database for liquid biopsy study (Aparicio et al, 2018, nucleic Acid Research); ctcRbase (http://www.origin-gene.cn/database/ctcRbase/), a gene expression database of liquid biopsy (Lei Zhao et al 2020, Batabase).

We added the following text in the second paragraph counting from the bottom on Page 3.

“After generation of raw sequencing data, applying the right bioinformatics algorithms is also crucial for mutation detection. A review that summarized bioinformatic tools for ctDNA analysis can be found in here [24]. There are also databases that can be used to help with the interpretation the liquid biopsy data. liqDB is a database for liquid biopsy small RNA sequencing profiles that provides users with meaningful information to guide their small RNA liquid biopsy research and to overcome technical and conceptual problems [25]. ctcRbase is a database that can be used for query and browse gene expression data of circulating tumor cells/microemboli [26].”

  1. Many companies made a kit or methods by panel sequencing, targeting some cancer-related genes. In this case, one could find novel somatic mutations. Please describe more how we could deal with novel somatic mutation or rare somatic mutation in terms of diagnosis.

Reply: We have added the discussion and examples on how to deal with de novo mutation in the last part of the first paragraph on Page 8. It goes “Although most of the ctDNA NGS assays focuses on detection of known mutations, it is also possible to identify de novo mutations. To validate if such de novo mutations are true mutations, references such as paired normal samples can be used to exclude normal variants [74]. The functions of such de novo mutations can be predicted by methods such as in silico prediction [75], thus may indicating if they are actionable or not. Because the mutation characteristics in the tumor and ctDNA are not static [76], thus continuous monitoring is important to ravel the status of disease.”Furthermore, readers could find references and databases mentioned in Question 2.

Reviewer 2 Report

Circulating tumor nucleic acids profiling with next-generation methods is a quickly developing area in clinical oncology. This review is summarized the most important knowledge about these technics.

I have some recommendations and comments.

The abstract and the conclusions sections are general. It is not clear in these sections what the purpose and the aim of the review were.

The numbering of the sections is inconsistent.

It would be worth mentioning cell-free RNAs including circulating tumor RNAs (ctRNA).

What is the relationship between tumor size and the amount of circulating tumor DNA (ctDNA)?

It is not clearly explained how the amount of ctDNA changes as a result of the oncology treatment. Is the yield of ctDNA suitable for monitoring the malignancy?

What is the difference between the mutant and the variant allele frequency (MAF vs. VAF)? What do the authors mean by mutation and variant?

There are studies to detect translocations/ gene fusions in ctDNA or ctRNA. I suggest adding these to the review.

It would be worth mentioning specific bioinformatics algorithms.

Whether there are targetable mutations in ctDNA reflecting mielo- and lymphoproliferative neoplasias (Table 1)?

Author Response

Circulating tumor nucleic acids profiling with next-generation methods is a quickly developing area in clinical oncology. This review is summarized the most important knowledge about these technics.I have some recommendations and comments.

Q1. The abstract and the conclusions sections are general. It is not clear in these sections what the purpose and the aim of the review were.

Reply: The purpose of this review was to fit the topic of “The Increasing Role of Next Generation Sequencing Methods in Mutation Analysis”. We focus on ctDNA profiling by NGS, which mainly relies on mutation analysis.

Q2. The numbering of the sections is inconsistent.

Reply: Thank you. We have corrected the wrong numeration.

Q3. It would be worth mentioning cell-free RNAs including circulating tumor RNAs (ctRNA).

Reply: ctRNAs are also part of the circulating nucleic acid that have great potential in tumor diagnosis and therapeutics. However, maybe due to their worse stability, there are less studies compared to ctDNA. It is still very important to remind the readers about their importance. Thus, we added some discussion between Page 2 and Page 3 in the last part of Section 2: “Circulating cell-free RNA (ccfRNA), mainly including mRNA and miRNA, is another type of cell-free nucleic acid that can be found in the bloodstream, they also possess the potentials to become biomarkers for cancer [15]. Although they share many similar characteristics as ctDNA, there are more challenges of using them as cancer biomarkers [16], thus more studies and technical advances are still needed before ccfRNA can become as popular as ctDNA in cancer diagnosis. Thus, we focus on the discussion of ctDNA in this review.” So that readers who are interested in ctRNA can refer to the related papers or reviews mentioned in here.

Q4. What is the relationship between tumor size and the amount of circulating tumor DNA (ctDNA)?

Reply: Indeed, readers may also be very curious about this. The relationship between tumor size and the amount of ctDNA seems to vary among different types of malignant carcinoma.  We therefore added “However, the amount of ctDNA is not simply associated with the tumor size, but varies in different types of cancer [11,12]” on Page 2 in the last paragraph.

Q5. It is not clearly explained how the amount of ctDNA changes as a result of the oncology treatment. Is the yield of ctDNA suitable for monitoring the malignancy?

Reply: In order to answer these two questions, we have elaborated Section 7 Detection of ctDNA mutations for cancer monitoring and assessment of minimal residual disease. Combined the answer from last question, we can conclude that the amount of ctDNA cannot be simply used for predicting the outcome of treatments. But the fraction of mutations can indeed reflect the status of disease and indicate the treatment efficacy. However, from the stand point of clinical observation, it’s still not clear that the amount of ctDNA is proportional to tumorigenesis either in progression or in after treatment

Q6. What is the difference between the mutant and the variant allele frequency (MAF vs. VAF)? What do the authors mean by mutation and variant?

Reply: We have now revised the descriptions of MAF and VAF to avoid more confusions. Please refer to the last paragraph on Page 3 where we now write “MAF or VAF describes the proportion of DNA molecules containing a mutation over the total number of molecules containing the same allele. Thus, they reflect the amount of ctDNAs when referring to the cfDNAs that contain tumor-specific mutant allele”.

Q7. There are studies to detect translocations/ gene fusions in ctDNA or ctRNA. I suggest adding these to the review.

Reply: We have added information about this. Now on Page 6 in the middle of Paragraph 1, it goes “as well as gene fusions and translocations that may be targetable in different types of cancer [47–50]”. We provided some references of this type of alterations. However, maybe due to the short sizes of ctDNA or ctRNA, this type of alterations is much less studied than mutations.

Q8. It would be worth mentioning specific bioinformatics algorithms.

Reply: Thank you for your suggestion. We have added a reference to guide readers to a review that summarize the bioinformatics algorithms and tools. We added “After generation of raw sequencing data, applying the right bioinformatics algorithms is also crucial for mutation detection in ctDNA data. A review that summarized classical and new bioinformatic tools for ctDNA analysis can be found in here [24]” on the second paragraph counting from the bottom on Page 3.  REF: Chen, S.; Liu, M.; Zhou, Y. Bioinformatics analysis for cell-free tumor DNA sequencing data. In Methods in Molecular Biology; 2018.

Moreover, We also introduce websites of liqDB (https://bioinfo5.ugr.es/liqdb), a small-RNAseq knowledge database for liquid biopsy study (Aparicio et al, 2018, nucleic Acid Research), ctcRbase (http://www.origin-gene.cn/database/ctcRbase/), a gene expression database of liquid biopsy (Lei Zhao et al 2020, Database) .

Q9. Whether there are targetable mutations in ctDNA reflecting myelo- and lymphoproliferative neoplasias (Table 1)?

Reply: Lymphoma diagnosis by ctDNA had been applied for near a decade, targetable mutation of ctDNA such as VDJ rearrangements and CD79B are commonly used. (Francesca Harrington, Mark Greenslade, Dipti Talaulikar, Greg Corboy Genomic characterization of diffuse large B-cell lymphoma Pathology (April 2021) 53(3), pp. 367–376; Rossi D, Spina V, Bruscaggin A, Gaidano G. Liquid biopsy in lym- phoma. Haematologica 2019; 104: 648 – 52.  Roschewski M, Staudt LM, Wilson WH. Dynamic monitoring of circulating tumor DNA in non-Hodgkin lymphoma. Blood 2016; 127: 3127 – 32.). We have added examples in Table 1.

Round 2

Reviewer 1 Report

Authors replied all the questions I raised. 

Reviewer 2 Report

The manuscript is acceptable for publication in the present form.